# Human Leukocyte Antigen and microRNAs as Key Orchestrators of Mild Cognitive Impairment and Alzheimer’s Disease: A Systematic Review

**DOI:** 10.3390/ijms25158544

**Published:** 2024-08-05

**Authors:** Cristina Sorina Cătană, Monica Mihaela Marta, Mădălina Văleanu, Lucia Dican, Cătălina Angela Crișan

**Affiliations:** 1Department of Medical Biochemistry, Faculty of Medicine, “Iuliu-Hațieganu” University of Medicine and Pharmacy, 400012 Cluj-Napoca, Romania; ccatana@umfcluj.ro; 2Department of Medical Education, Faculty of Medicine, “Iuliu-Hațieganu” University of Medicine and Pharmacy, 400012 Cluj-Napoca, Romania; mmarta@umfcluj.ro; 3Department of Medical Informatics and Biostatistics, Faculty of Medicine, “Iuliu-Hațieganu” University of Medicine and Pharmacy, 400012 Cluj-Napoca, Romania; mvaleanu@umfcluj.ro; 4Clinical Institute of Urology and Renal Transplantation, 400000 Cluj-Napoca, Romania; 5Department of Neurosciences, Faculty of Medicine, “Iuliu-Hațieganu” University of Medicine and Pharmacy, 400012 Cluj-Napoca, Romania; ccrisan@umfcluj.ro

**Keywords:** Alzheimer’s disease, mild cognitive impairment, human leukocyte antigens, microRNAs, biomarkers, neuroinflammation

## Abstract

The expression of inflamma-miRs and human leukocyte antigen (*HLA*) haplotypes could indicate mild cognitive impairment (MCI) and Alzheimer’s disease (AD). We used international databases to conduct a systematic review of studies on *HLA* variants and a meta-analysis of research on microRNAs (miRNAs). We aimed to analyze the discriminative value of HLA variants and miRNAs in MCI, AD and controls to evaluate the protective or causative effect of HLA in cognitive decline, establish the role of miRNAs as biomarkers for the early detection of AD, and find a possible link between miRNAs and *HLA*. Statistical analysis was conducted using Comprehensive Meta-analysis software, version 2.2.050 (Biostat Inc., Englewood, NJ, USA). The effect sizes were estimated by the logarithm base 2 of the fold change. The systematic review revealed that some *HLA* variants, such as *HLA-B*4402*, *HLA-A*33:01*, *HLA-A*33:01*, *HLA-DPB1*, *HLA-DR15*, *HLA-DQB1*03:03*, *HLA-DQB1*06:01*, *HLA-DQB1*03:01*, SNPs on *HLA-DRB1/DQB1*, and *HLA-DQA1,* predisposed to cognitive decline before the occurrence of AD, while *HLA-A1*01*, *HLA-DRB1∗13:02*, *HLA-DRB1*04:04,* and *HLA-DRB1*04:01* demonstrated a protective role. The meta-analysis identified let-7 and miR-15/16 as biomarkers for the early detection of AD. The association between these two miRNA families and the *HLA* variants that predispose to AD could be used for the early screening and prevention of MCI.

## 1. Introduction

Age-related cognitive decline remains a major medical and social problem. The aging phenotype is characterized by increased cellular senescence, reduced stem cell population, and altered proteostasis, which activate the inflammasome, i.e., the multiprotein oligomer responsible for inflammation, change in intercellular communication, and loss of telomere function [1]. The entire aging-associated pathology, known as age-related diseases (ARDs), which include Alzheimer’s disease (AD), cancer, atherosclerosis, and metabolic diseases, is coordinated by increased levels of pro-inflammatory cytokines through the nuclear factor-kappa B transcription factor (NF-kB) pathway [2,3].

Statistics show that more than 16 million people in the United States are living with cognitive impairment and 6.7 million have AD [4]. According to the World Health Organization (WHO), 55 million people have dementia, with 7.7 million new cases occurring every year worldwide. The prevalence of dementia, particularly AD, is increasing fast in both developed and developing countries [5]. The worldwide prevalence of persons at risk for dementia or cognitive impairment due to AD was estimated at 315 million [6]. By 2050, it is expected that one in 85 people will be affected by this pathological condition worldwide, especially since the evolution of patients with mild cognitive impairment (MCI) toward dementia can now be accurately predicted using deep learning models and changes in dynamic magnetic resonance imaging (MRI) markers [7]. A medial temporal lobe atrophy on MRI can differentiate between healthy aging and AD, with such information being included in new research diagnostic criteria for AD, prodromal AD, and MCI due to AD. MRI also has more than 80% sensitivity and specificity in differentiating AD from vascular dementia or dementia with Lewy bodies, at the same time predicting the progression from MCI to AD with almost the same level of accuracy [8].

The concept of MCI or the pre-dementia stage is highly significant in the field of aging, since MCI is regarded as a borderline condition between normal aging and very early dementia. Individuals with MCI have a high risk of developing dementia and higher mortality rates compared to cognitively normal people [9]. Depending on the cause, patients with MCI remain stable, return to normal, do not develop AD, or develop AD [10]. Due to the scarcity of disease-modifying treatments for dementia, the importance of diagnosing and initiating early treatment during the MCI stage has been widely recognized as a key strategy for the effective management of this condition, with early intervention offering the potential for improved long-term outcomes. 

The rate of progression from mild MCI to AD is estimated at 10% per year and can reach up to 80–90% after six years [11]. The identification of individuals at a high risk of developing AD before the occurrence of cognitive decline is crucial so that such patients can benefit from curative interventions. The methods and procedures for diagnosing MCI, which combine neuropsychological assessment, biomarkers, and neuroimaging techniques, have already been applied in clinical practice [12], although an accurate diagnosis of central nervous system (CNS) pathology may be costly, invasive, and potentially dangerous [13]. Some of the limitations of the currently available techniques include the inability to identify, with high sensitivity, individuals who are in the symptomatic pre-dementia stage of AD, or who are pre-symptomatic but at a high risk of experiencing the clinical onset of MCI, as well as measuring and tracking very subtle cognitive changes over time in these same individuals [14,15,16]. The older criteria for MCI diagnosis were regarded as incomplete when research concluded that not all the patients with MCI developed dementia and that memory impairment was not the only cognitive domain altered [10]. Therefore, a more comprehensive definition of MCI was required to facilitate the diagnosis and monitoring of MCI patients [10,17].

MCI can be classified according to the category of the cognitive domain affected: learning and memory, complex attention, social cognition, executive function, visuospatial function, and language. MCI is regarded as non-amnestic when the patient’s memory is relatively intact, although progression to AD dementia is possible; and amnestic, which is the more common form characterized by deficits in one or several cognitive domains, predominantly memory, and a high risk of progression to AD. According to a comprehensive neuropsychological classification system implemented in 2009, MCI can be amnestic, when it impairs recall and recognition; mixed, when it affects domains such as language, executive function, recall and recognition, or visuospatial functioning; dysexecutive, if it impairs attention, executive, and visuospatial functions, but the memory remains intact; and visuospatial, if only one measure of visual construction is affected [10,17,18]. The phenotypic classification of MCI with clinical information and laboratory tests can currently contribute to establishing the probable cause of MCI and predict its evolution. For instance, patients with dysexecutive MCI (dMCI) are less likely to progress to Alzheimer’s- type dementia but more likely to have a stroke [10].

A better characterization of the intermediate stage between normal aging and the diagnosis of clinically probable very early AD may contribute to improved diagnostic mechanisms, including fluid biomarkers and timely therapeutic measures for MCI [19,20,21]. The aging society and increased life expectancy are the strongest risk factors for MCI. However, some protective factors, which are associated with a reduced incidence of dementia or delayed dementia onset, may also protect against MCI. Thus, higher education, bilingualism, and cognitively stimulating activities protect cognition, and lifestyle factors such as the Mediterranean diet, physical activity, smoking cessation, mild-to-moderate alcohol consumption, and participation in social activities are associated with a reduced risk of cognitive impairment [10,18]. All these protective factors increase the cognitive reserve, thus improving tolerance to more neuropathologies without cognitive and functional decline and slowing down the development of dementia. The cognitive reserve is influenced by the anatomical substrate of the brain or the adaptability of cognition, depending on the above-mentioned factors [22,23,24].

As aging progresses, the chronic inflammatory process known, as age-related diseases (ARDs), which include tumors, dyslipidemia, hypertension, and oxidative stress, are more likely to occur [2] due to the potential involvement of mechanisms that include neuroinflammation, synaptic dysfunction, epigenetic modifications, oxidative stress responses, proteasomal impairment, and abnormal immune responses [25].

## 2. Inflammatory microRNAs as New Players in MCI and AD

The discovery of microRNAs (miRNAs) revealed another level of genetic and epigenetic alterations involved in cognitive decline. Besides its impact on cognitive functions, AD is characterized by extracellular amyloid plaques composed of the amyloid-β peptide (Aβ), loss of synapses and intracellular aggregates of hyperphosphorylated Tau protein. MicroRNA dysregulation was repeatedly reported in association with key genes that regulate Aβ synthesis, cleavage, and clearance. The modulatory role of miRNAs in cognitive decline covers at least three areas: Tau pathology (phosphorylation induced by miR-125, miR-138, miR-146 and aggregation by the protective miR-132, miR-369, miR-483-5p, miR-181c, miR-212), neuroinflammation (induced by miR-155, miR-34a, miR-181a), and Aβ disruption (production and metabolism by protective miRNAs such as miR-200, miR-137, miR-15b, miR-98, and miR-101, and clearance by miR-34a and miR-29b) [26]. Similar to an inflammatory process, the expression in blood of inflamma-miRNAs such as miR-126-5p, miR-29a, and miR-125b could be indicative of AD progression, thus becoming diagnostic biomarkers of AD. In addition, inflamma-miRNAs may also be promising therapeutic targets in AD patients [27].

High-throughput technologies currently enable miRNA profiling studies able to identify miRNA differential expression in cognitively healthy individuals and persons who are experiencing very subtle cognitive changes that may signal the early onset of MCI. Over 70% of the identified miRNAs are expressed in the human brain and have a crucial role in multiple inflammatory processes, so they can be regarded as promising blood biomarkers for MCI [11,28]. 

miRNAs also have the advantage of being stable in bodily fluids and less invasive, thus contributing positively to patient care and outcomes. Persistent inflammation is an omnipresent feature of the aging process and most ARDs. Research has recently shown that the spectrum of miRNAs is highly specific to different pathologies, which contributes to distinct patterns of gene expression. The fact that a single miRNA has multiple targets is crucial for understanding the role of miRNAs in normal and pathological processes. The target mRNAs of a given miRNA could be predicted using homologies between the miRNA seed region and the complementary site of the target mRNA 3′-untranslated region (UTR) [28,29]. 

The discovery of extracellular miRNAs enabled the use of small non-coding molecules for monitoring the biomarkers of aging. Until 2018, there was no agreement on whether miRNAs were the key players in aging. Researchers were particularly interested in the inflammatory response and found that a few miRNAs had an important role in this respect. They also showed that some small non-coding RNAs modulate inflammation. Three of them, namely miR-21, miR-126, and miR-146a, as well as their target mRNAs belong to the NF-kB pathway, which is the master modulator of the pro-inflammatory status in ARDs [28]. Similarly, our preliminary studies revealed that inflamma-miRNAs modulate the NF-kB signaling pathway manifested by increased levels of pro-inflammatory cytokines, thus lying at the intersection between aging and inflammation [28,30]. 

Besides these results, our previous findings also showed that miRNA profiling seems to be involved in hippocampus-dependent functions, such as learning or episodic and working memory. The role of miRNAs in aging was demonstrated through their relationship with the learning and memory function. The inhibition of miR-124 resulted in an enhanced capacity of spatial learning and working memory. Conversely, miR-9 was found to support the capacity of spatial learning [3,27,30]. 

Moreover, miR-9-5p alleviated Aβ-induced synaptotoxic impairment by inhibiting calcium/calmodulin-dependent protein kinase kinase 2 (CAMKK2); it decreased the clearance of Tau proteins by targeting ubiquitin conjugating factor E4 B (UBE4B), and it was related to the amyloid cascade hypothesis and memory loss by targeting several transcription factors, such as β-site amyloid precursor protein-cleaving enzyme 1 (BACE1), silent information-regulator sirtuin 1 (SIRT1), and CAMKK2 [31,32,33]. Therefore, we confirmed the links between the pro-inflammatory mechanism underlying ARDs and the precise function of certain miRNAs in cellular senescence (CS) [29,30].

The identification of miRNA profiles in cognitive impairment is an important step toward MCI diagnosis and prognosis. At the same time, the extracellular nature of the transport mechanisms of miRNAs allows for the measurement of genetic material from the central nervous system through bodily fluids, such as saliva, blood, and serum. Recent studies on saliva-based miRNAs in autism disorders, correlated with adaptive behavior, provided compelling evidence for the addition of miRNA biomarker screening to the diagnosis of MCI [34,35].

## 3. Human Leukocyte Antigen (HLA) Genetic Variants in MCI and AD

Another area of genetic research has focused on the effects of HLA genetic variants on MCI and AD. The polygenic major histocompatibility complexes I and II (MHC-I and MHC-II), known as the HLA complexes, are glycoproteins that code for peptides found on the surface of antigen-presenting cells (APCs); thus, HLA discriminates between self and non-self T lymphocytes. Both classes of proteins share the same binding platform, a functional trimeric complex composed of two domains originating from a single heavy α-chain (HC) in the case of MHC class I and two chains in the case of MHC class II (α- and β-chains). As far as HLA class II is concerned, one immunoglobulin (Ig) domain is present in each chain, while the second Ig-type domain of HLA class I is represented by the non-covalent association between β_2_-microglobulin (β_2_M), which is a short invariable light chain subunit, and HLA class I molecules on the surface of cytotoxic CD8+T cells. Moreover, transmembrane helices anchor the HC of HLA class I and both HLA class II chains in the cell membrane [36].

The genomic map of the HLA gene complex revealed that the *HLA* class I gene complex corresponds to the genes coding for the HLA-A, HLA-B, and HLA-C molecules, while the *HLA* class II region encodes for HLA-DR, HLA-DQ, and HLA-DP. In addition, there are many non-classical *HLA* class I genes, including MHC class I chain-related proteins A and B (MICA and MICB), HLA-E, HLA-G, HLA-F, and the human homeostatic iron regulator protein (HFE). There are also two genes encoding the transporter associated with Ag processing (TAP), which is a heterodimeric member of the ATP-binding cassette transporter family located in the HLA class II region close to the immunoproteasome genes (Lmp2 and Lmp7). The TAP-binding protein (*TAPBP*) gene is situated at the edge of the HLA locus. LMP 2, LMP 7, TAP-1, and TAP-2 are involved in the processing and transport of peptides presented by both types of HLA molecules [37,38].

A third class of genes, which includes genes for tumor necrosis factors (TNFs), complement components such as C2 and C4, 21-hydroxylase and others, is located between the class I and II regions [37]. The *HLA* gene variants within MHC class I and II regions associated with cognitive decline in Alzheimer’s disease (AD) include *HLA* class I genes (*HLA-A2*, *HLA-A1*, *HLA-A24*, *HLA-B*4402*, *HFE* H63D, *HFE* C282Y), *HLA* class II genes (*HLA-DQB1*06* (*HLA-DQ*), rs9271192, *HLA-DRB1*15*, *HLA-DRB1*04*, *HLA-DR15* haplotype (HLA-DR)), and TAP2 SNP rs241448 [38]. 

The neuronal expression of MHC-I has recently been shown to control synaptic plasticity and neurite outgrowth. MHC-I is destabilized in the brains of AD patients and neuronal cells treated with oligomeric β-amyloid (Aβ) [39]. 

Fine-mapping of the human leukocyte antigen (HLA) region highlighted the neurological and immune-mediated disease haplotype *HLA-DR15* as a risk factor for late onset Alzheimer’s disease (LOAD) when symptoms occur after the age of 65 [40]. In addition to mutations in the most important causative genes for familial AD (FAD), such as amyloid precursor protein and presenilins, recent research has identified other genes with a possible contribution to the progression of FAD. Such genes, involved in inflammation, cholesterol metabolism, and the innate immune system of the brain, include clusterin (*CLU*), ATP-binding cassette transporter subfamily A member 7 (*ABCA7*), bridging integrator 1 (*BIN1*), triggering receptor expressed on myeloid cells 2 (*TREM2*), sortilin-related receptor-1 (*SORL1*), phosphatidylinositol-binding clathrin assembly protein (*PICALM*), CD33CD2-associated protein (*CD2AP*), complement component (3b/4b) receptor 1 (*CR1*), and phospholipase D3 (*PLD3*) [41]. 

*HLA-DRB1/DQB1* gene variants modulate the susceptibility of AD [42]. Statistical evidence supports the existence of a real link between the *HLA-DRB1* polymorphism and LOAD, thus revealing that carriers of C allele at rs9271192 present an increased risk of developing short- and long-term memory loss due to late-onset dementia [43].

There is a paucity of data on the localization, expression, and aberrant immune regulation in AD physiopathology of the HLA pathway, which functions pleiotropically in the CNS of both males and females with cognitive decline. HLA has multiple functions in the CNS, such as synaptic plasticity and refinement, thus participating in activity-dependent structural remodeling processes. Currently available studies characterized the cellular expression of HLA in the human brain as a result of DNA promoter methylation, leading to the different gene expression of the HLA class I complex according to neural regions, ages, and genders. Under certain conditions, HLA-I reduces glutamatergic and GABAergic synapse density, and inhibits synaptic plasticity, thus being a potential pathway of brain aging. Additionally, genetic variations modify HLA charge distribution, hydrophobicity, geometry, peptide interactions, and the severity of certain diseases [44,45]. 

The challenges of finding suitable case control and cohort studies for a systematic review include ethnic diversity, the organization of the HLA region, the numerous sequencing methods available, as well as the non-uniform sensitivity and specificity of the statistical parameters used to quantify the diagnostic and prognostic potential of HLA variants in AD. Starting from these limitations, our workflow aimed to discriminate between AD and normal controls (NC) according to the HLA haplotypes.

## 4. Systematic Review Methods

The article was reported according to PRISMA, and the research protocol is available at INPLASY–International Platform of Registered Systematic Review and Meta-analysis Protocols https://inplasy.com/ (accessed on 11 July 2024) (registration number INPLASY202470045; DOI number 10.37766/inplasy2024.7.0045).

### 4.1. Information Sources

For the systematic review, PubMed and Web of Science were searched for case control and cohort studies published in English between January 2015 and June 2024.

### 4.2. Search Strategy

We used the following keywords: Alzheimer’s disease, mild cognitive impairment and HLA, diagnostic, biomarker, and haplotype to find relevant articles. We previously identified let-7 and miR-15/16 as biomarkers for the early detection of AD [1]. Therefore, the following keywords were used to search for studies published in English in Medline and EMBASE: neurodegenerative disease, Alzheimer’s disease, microRNAs and diagnosis, biomarkers, and microRNA profiling. The PubMed search strategy was the following: “((“alzheimer disease” [MeSH Terms] OR (“alzheimer” [All Fields] AND “disease” [All Fields]) OR “alzheimer disease” [All Fields] OR (“alzheimer s” [All Fields] AND “disease” [All Fields]) OR “alzheimer s disease” [All Fields]) AND (“cognitive dysfunction” [MeSH Terms] OR (“cognitive” [All Fields] AND “dysfunction” [All Fields]) OR “cognitive dysfunction” [All Fields] OR (“mild” [All Fields] AND “cognitive” [All Fields] AND “impairment” [All Fields]) OR “mild cognitive impairment” [All Fields]) AND (“hla” [Title/Abstract] OR “hla” [All Fields]) AND (“diagnosis” [MeSH Terms] OR “diagnosis” [All Fields] OR “diagnostic” [All Fields] OR “diagnostical” [All Fields] OR “diagnostically” [All Fields] OR “diagnostics” [All Fields]) AND (“biomarker s” [All Fields] OR “biomarkers” [MeSH Terms] OR “biomarkers” [All Fields] OR “biomarker” [All Fields]) AND “haplotype” [All Fields])”. For neurodegenerative disease the following search strategy was added: “(“neurodegenerative diseases” [MeSH Terms] OR (“neurodegenerative” [All Fields] AND “diseases” [All Fields]) OR “neurodegenerative diseases” [All Fields] OR (“neurodegenerative” [All Fields] AND “disease” [All Fields]) OR “neurodegenerative disease” [All Fields])”.

### 4.3. Eligibility Criteria

The inclusion criteria included original articles on HLA haplotypesin AD patients compared with the controls. We included the case control and cohort studies of patients with AD validated with Mini Mental State Examination and tested for HLA subtypes. Abstracts, reviews and studies with incomplete data were excluded. Relevant systematic reviews were examined for relevant data sources. The selected studies included human patients with AD validated by magnetic resonance imaging (MRI) or the Mini Mental State Examination (MMSE).

### 4.4. Selection Process

Two authors (CSC and LD) screened the titles and abstracts of the identified articles. In case of disagreement, the articles were selected by discussion. The same two authors assessed the full text of the selected articles. Here too, in case of disagreement, the articles were selected by discussion. 

### 4.5. Data Collection Process and Data Items

For each study, data on HLA class, effects on cognitive impairment, and HLA frequencies were extracted. The main outcomes are represented by fold change.

## 5. Meta-Analysis Methods

Due to the heterogeneity of the results, out of the 74 studies initially found, we selected only 8 studies that used fold change as the effect measure. Statistical analysis was conducted using version 2.2.050 of Comprehensive Meta-analysis software and the effect sizes were estimated by the logarithm base 2 of the fold change (FC). The up-regulated miRNAs correspond to positive logarithmic values, while the down-regulated non-coding RNAs correspond to negative logarithmic values. Forrest plots were used to present the results. 

## 6. Results 

For the systematic review, we identified 184 potentially relevant studies based on an electronic search in PubMed and Web of Science. After thoroughly analyzing all these studies, we selected 32 studies that presented the link between cognitive ability and HLA class I and II (Figure 1). The genetic association between HLA and cognitive impairment from AD is presented in Table 1.
ijms-25-08544-t001_Table 1Table 1Selected studies for the overall discrimination between AD and NC.HLA TypeEffects on Cognitive ImpairmentReferencesHLA class II*HLA-DR15* haplotype−causes neuroinflammation[40]−carriers showed cognitive decline over time[46]−risk factor for dementia[47]−decline in the function of the adaptive immune system in AD[48]*TAP2* rs241448*HLA*-*DRB1* (linkage disequilibrium) in Caucasian populations−increase susceptibility to AD by facilitating the access of herpes simplex virus type 1(HSV-1) to the brain[49,50,51]*HLA-DQB1*06:01*(Asian populations)−modulates the atrophy of the left posterior cingulate volume[42,52,53]*HLA-DQB1*03:01* (all populations)−increases susceptibility to AD[42,52,53]Single-nucleotide polymorphisms (SNP) on *HLA-DRB1/DQB1*:rs9271192, rs35445101, rs1130399, rs2854275, and rs28746809−cause neurodegeneration[42,43,52,54,55,56]*HLA-DRB1*09:01*−causes neuroinflammation−facilitates the alteration of the left posterior cingulate region−increases susceptibility to AD[42,52,54,57,58,59,60]*HLA-DRB1**∗**13:02*−protect against AD and neural network age-related deterioration[59]*HLA-DRB1*04:04*[61]*HLA-DRB1*04:01*[62]*HLA-DQA1*−causes neuroinflammation−is overexpressed in microglia−is associated with immune activation and inflammation−carries increased risk of developing AD[57,60,62,63,64,65]HLA class I*HLA-B**HLA-B*4402*−cause brain atrophy−produce cognitive decline[66,67,68]*HLA-A2**HLA-A*33:01*−maintain synapses−have stimulatory effect on the risk that depends on genetic loadings−establish a link between neurodegenerative and immune processes−alter hippocampal volume[42,69,70]*HLA-A1**HLA-A1*01*−delay AD development[69,71,72]Homeostatic iron regulator (*HFE*) (HLA-H)-C282Y*HFE* (HLA-H)-H63D mutation(high iron)−cause synaptic dysfunction−increase Tau phosphorylation[73,74,75,76,77]*HLA-DPB1*−causes neurodegeneration[59]


For the meta-analysis, out of the 74 possibly relevant studies, 31 provided targeted data about the miRNAs involved in AD, but only 8 used the fold change as statistical indicators, with the other 22 therefore being excluded. The results were presented as two forest plots of up- and down-regulated microRNAs in AD patients [78]. 

The results also revealed the encouraging role of miRNAs, especially the let-7 (8/12 members) and miR-15/16 (4 members) families (Figure 2). These biomarkers used in the diagnosis, monitoring, and early detection of AD have the advantage of being low-cost and non-invasive. We previously confirmed the diagnostic value of miRNAs expressed in different body fluids by analyzing the discriminative value of miRNAs in both groups (control subjects versus AD patients) [78]. MCI is known to have several outcomes, since patients can remain stable, revert to normal or progress to AD. Our current systematic review identified that some HLA variants predispose to cognitive decline before the occurrence of AD.

The let-7 family, containing 8–12 members and miRNAs-15/16 with 4 members, was found to be dysregulated in AD compared to normal controls. The members were up- or down-regulated, and had a significant impact in the progression to AD [1]. The target population includes pre-symptomatic and symptomatic MCI individuals who could undergo early screening to assess the conversion from amnestic MCI to AD besides neuropsychiatric tests such as the clock drawing test (CDT) and Mini Mental State Examination (MMSE). Thus, such predictive biomarkers are part of personalized prevention if the target population is screened early using biological fluids such as saliva, which is a non-invasive procedure. Therefore, future studies should establish which intrinsic or extrinsic external factors can influence the reversion of MCI to normal cognition, indicated by improved patterns in the execution of the CDT. 

## 7. Discussion

Our systematic review revealed that some *HLA* variants predispose to cognitive decline before the occurrence of AD. These variants are *HLA* class II: *HLA-DR15* haplotype, *HLA-DQB1*03:03*; *HLA-DQB1*06:01* (Asian populations), *HLA-DQB1*03:01* (all populations), SNPs on *HLA-DRB1/DQB1* (rs9271192, rs35445101, rs1130399, rs2854275, and rs28746809), *HLA-DQA1;* and HLA class I: *HLA-B*4402*, *HLA-A*33:01*, *HLA-A*33:01*, and *HLA-DPB1*. Conversely, other HLAs, such as *HLA-A1*01*, *HLA-DRB1∗13:02 HLA-DRB1*04:04* or *HLA-DRB1*04:01,* protect against age-related cognitive deterioration.

Besides the above-mentioned HLAs, there is also HLA-G, a non-classical member of HLA class I, known for its immunomodulatory properties, which could be a crucial next-generation immune checkpoint in cancer and other diseases. Since HLA-G can be shed from the cell surface or released by various cells as free soluble HLA-G (sHLA-G) or as part of extracellular vesicles (EVs), namely HLA-G-bearing EVs (HLA-G_EV_), the potential of sHLA-G and HLA-G_EV_ as predictive biomarkers was studied in various types of inflammation [79].

Pro-inflammatory responses were also described for HLA-G molecules, the HLA-G homodimer being shown to induce the secretion of the pro-inflammatory cytokines interleukin-6 and -8, and the tumor necrosis factor alpha from both decidual macrophages and natural killer cells [80].

Chronic inflammation was also validated through modified expression levels of miRNAs in different stages of AD development, as well as in different brain regions. For example, inflamma-miRs, such as miR-34c, miR-146a-5p, and miR-16, were up-regulated in early AD, while miR-107, miR-128a, miR-16, and miR-146a-5p were down-regulated in late AD [81].

Similarly, our meta-analysis revealed the encouraging roles of miRNAs in the diagnosis of AD, especially let-7 (8/12 members) and miR-15/16 (4 members) families, which were both up- and down-regulated. However, the heterogeneity of miRNA expression in the hippocampus, cerebrospinal fluid and peripheral blood, as well as the small sample size of the studies and the different methods for miRNA detection were the main obstacles in interpreting the results.

Establishing interactions between the HLA and miRNA families identified by our systematic review and meta-analysis is crucial for future miRNA-based therapeutics in MCI and AD. For example, the interaction between *HLA-DRB1* and miR-3928 was established in inflammatory diseases such as rheumatoid arthritis [82].

A previous study highlighted that miR-15, 16, and miR-744 had opposing roles in modulating classical and non-classical MHC class I molecules by targeting the coding sequence (CDS) in other neurological disorders. It also demonstrated that MHC class I regulation showed, for the first time, miRNA-dependent control mediated by the CDS. CDS-located miRNA binding sites could improve the general use of miRNA-based therapeutic approaches, as these sites are highly independent of structural variations (e.g., mutations) in the gene body [83]. microRNAs (miRNAs) are known to be essential for controlling gene expression, their deregulation being associated with the development and progression of various diseases, including AD. In this respect, a discordant messenger RNA/protein expression leading to extensive post-transcriptional regulation of MHC class I molecules was already revealed by our systematic review. Unfortunately, only a very limited number of miRNAs that target these molecules was discovered [83]. 

Another interaction between *HLA-DPB1*, which, according to our systematic review is involved in neurodegeneration, and miR-let-7b-5p, which is known to prevent AD progression, was previously identified in other studies focusing on inflammation in sepsis. *HLA-DPB1* and transcription factor Spi-B were identified as promising targets for the miR-let-7 family, because they play a key role in maintaining the pluripotency of endogenous stem cells, which is essential in different medical conditions [84,85]. 

A recent study used the precise statistical analysis of RNA data derived from the blood samples of a human cohort population to demonstrate, for the first time, that many *HLA* and non-HLA genes (multilocus expression units) and splicing mechanisms were regulated by eight structurally polymorphic SINE-VNTR-Alu (SVAs) within the MHC genomic region. Thus, SVAs within the MHC region were shown to be important regulators or rheostats of gene co-expression, with potential roles in diversity, health, and disease [86]. R_SVA_85 and NR_SVA_381 were found to have regulatory effects on the expression of *HLA-DPA1* and *-DPB1* transcription. A homozygous R_SVA_85 insertion (PP) increased the transcription of *HLA-DPB1,* whereas a homologous NR_SVA_381 insertion (PP) decreased the transcription levels of *HLA-DPA1* and *HLA-DPB1* in a Parkinson’s Progression Markers Initiative (PPMI) cohort [87].

The currently available research has revealed that certain HLA haplotypes and microRNAs can be used to select populations at risk of developing cognitive decline. Salivary screening for miR-15, 16 and let-7 families, which are dysregulated in AD, facilitates the selection process. Thus, efficient measures can be taken to increase the cognitive reserve and stimulate reversion rates from MCI to normal cognition, which prevents progression to dementia. MCI assessment and genetic analysis represent relatively easy and low-cost screening methods that could be used to avoid the progression of amnestic MCI to AD. 

### Limitations

There are several limitations in the evidence included in the review. One of the most important is the inability to identify, with high sensitivity, people who are in the pre-dementia stage of AD or who are pre-symptomatic but at a high risk of developing the clinical onset of MCI, as well as the inability to measure and monitor extremely subtle changes in cognition over time in these same people. There is an important clinical heterogeneity in the stage of the disease: HLA types and haplotypes. Concerning the review process, only two databases were used to identify the available literature. 

The findings of this systematic review suggest that specific HLA variants can be used as biomarkers for the early identification of individuals at a risk of cognitive decline, allowing for timely interventions. Incorporating HLA genotyping into clinical assessments could facilitate personalized management strategies for MCI and AD. The identified protective HLA variants provide a basis for risk stratification and tailored preventive measures. miRNAs have potential as non-invasive diagnostic tools for early AD detection, despite challenges in expression heterogeneity and detection methods. This knowledge encourages the development of precise and early diagnostic tests and personalized treatment plans.

## 8. Conclusions

The identification of a network of interactions between different exogenous miRNAs and the HLA complex could also have beneficial effects in minor cognitive impairment by preventing irreversible AD dementia. In-depth studies are further required to establish how HLA control can be modulated by miRNAs. This would provide protective effects in the early stages of cognitive decline and thus avoid the occurrence of the age-related pathology affecting millions of people worldwide.

## Figures and Tables

**Figure 1 ijms-25-08544-f001:**
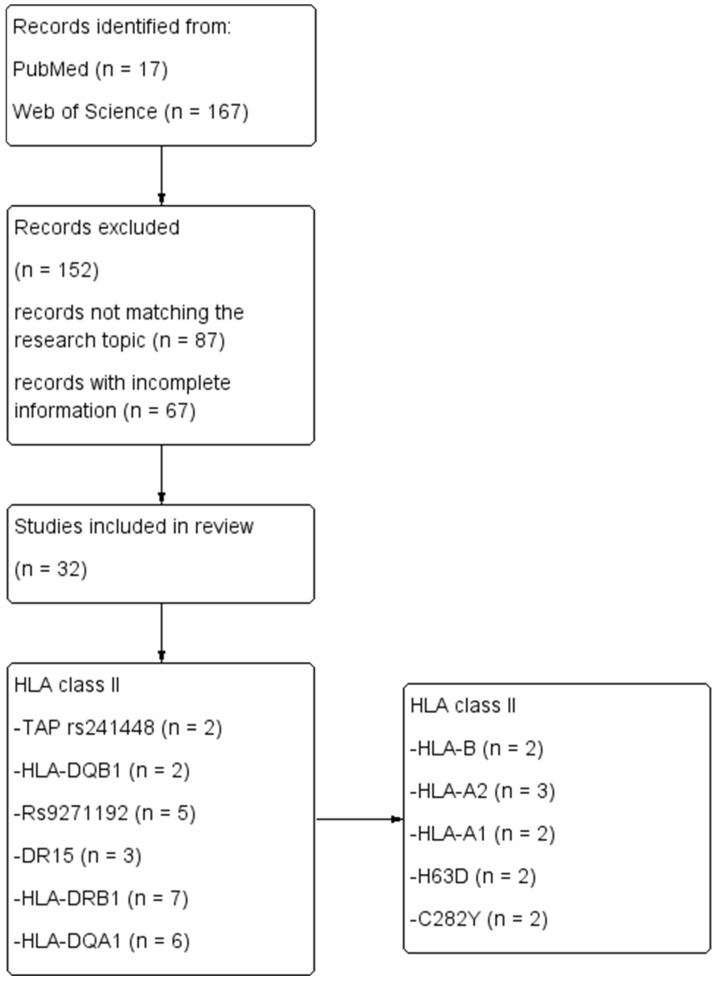
Prisma flow diagram of the selection criteria for the HLA systematic review.

**Figure 2 ijms-25-08544-f002:**
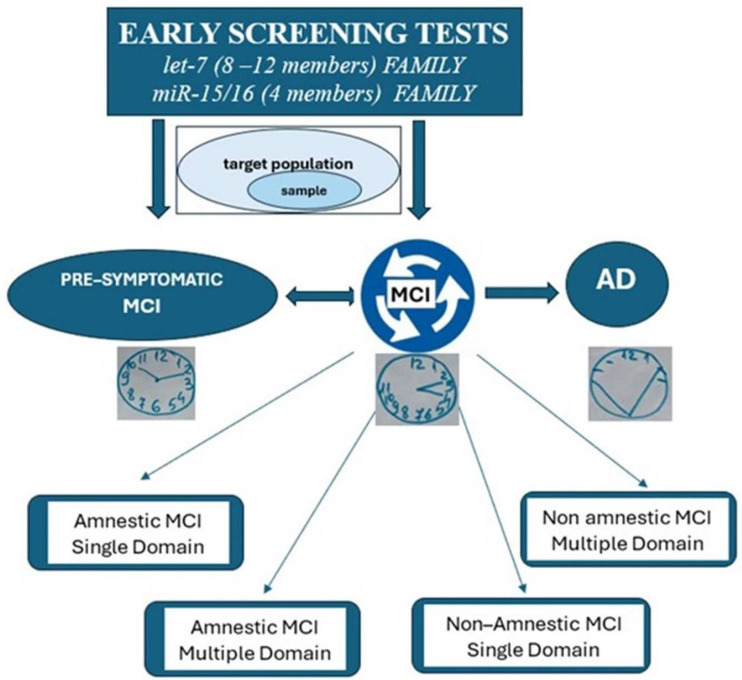
Potential diagnostic value of miRNAs-15/16 and let-7 in pre-symptomatic and symptomatic MCI.

## Data Availability

Not applicable.

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
