# Peer review of "Human Leukocyte Antigen and microRNAs as Key Orchestrators of Mild Cognitive Impairment and Alzheimer’s Disease: A Systematic Review"

_ijms, 2024, doi:10.3390/ijms25158544_

Round 1

Reviewer 1 Report

Comments and Suggestions for Authors

The authors have presented a systematic review of the role of HLA and miRNAs in AD and cognitive dysfunction and its comparison to control. The review is worth investigating as AD lacks a concrete therapeutic approach. I have a few comments.

·      The Introduction part mainly gives background about MCI, AD, and dementia with lesser emphasis on the core hypothesis of the review.

·      Did the authors go through full text during the screening of articles or it was done based on the title and abstract as well?

·      What kind of articles were included? Was it exclusively research articles? Preclinical or clinical?

·      How these miRNAs are relevant to major pathological hallmarks amyloid plaques and neurofibrillary tangles of AD? Can it be explained?

Comments on the Quality of English Language

Minor editing is required.

Author Response

For Reviewer 1:

Thank you very much for taking the time to review our manuscript and provide us with
encouraging feedback. We appreciate your support.
Based on your valuable observations, we added information on major pathological hallmarks
such as amyloid plaques and neurofibrillary tangles in AD – please see lines 120-129.
We clarified the inclusion criteria – please see lines 279-286.
The selection process is described in lines 288-291.
We also performed English language editing (lines 249-293 and 426-436)
We appreciate your useful insights, and we hope that the revised version of the manuscript is
clearer and more informative.

For Reviewer 2:

Thank you very much for taking the time to review our manuscript and provide us with valuable
suggestions. We revised the manuscript and included information on the expression and
regulation of HLA genes according to the two suggested sources (please see lines 401-411),
which were also added to the References list (lines660-663).
We appreciate your useful insights, and we hope that the revised version of the manuscript is
clearer and more informative.

Reviewer 2 Report

Comments and Suggestions for Authors

I had only minor queries regarding the poor quality of figs 2-3, the missing use of international rules for the names of genes and the high ithenticate score (38%).

Comments on the Quality of English Language

-

Author Response

For Reviewer 3:
Thank you very much for taking the time to review our manuscript. Based on your valuable
suggestions, we deleted Figures 2 and 3 and revised the names of genes. We also asked our
university library to check the manuscript using an official anti-plagiarism software used in our
country. The results showed similarity percentages of 12.26% (for minimum 5 words) and 4.17%
(for minimum 25 words), respectively. The report is available at:
https://lmsapi.plagiat.pl/report/?language=ro&shareId=d20de9f0-ee48-432b-8cf0-5eda1cb7475b
We appreciate your useful insights, and we hope that the revised version of the manuscript is
clearer. 

Reviewer 3 Report

Comments and Suggestions for Authors

Tghuis is a very interetsing manuscript.

But maybe the authors can elaborate more on the expression and regulation of the HLA genes. Here are two recent papers (PMID: 38590523, 38031415) that have addressed this question very well and provide strong evidence on the HLA gene expression and regulation and its involvement in Parkinson's disease. This is relevant to the present manuscript.

- The main question addressed by the research is the expression of HLA genes and their relation to AD.

-All parts are original and relevant.

-The expression of HLA genes, specific subtypes of the genes.

-New references need to be added that show how HLA genes are regulated and related to Parkinon’s disease.

-Conclusions are supported by the evidence.

-The references appropriate are generally, but two references are missing.

Author Response

(The authors gave the same response as above.)
